# FLOW CURVATURE EXPLAINS FAILED SDE DRIFT ESTIMATION UNDER SPARSE SAMPLING

**Dimitra Maoutsa** [*]
Technical University of Berlin
dimitra.maoutsa@gmail.com

## ABSTRACT

Drift estimation from sparsely observed stochastic trajectories usually performs poorly as the sampling interval $\tau$ grows, even for moderate dynamical noise. A common probabilistic account attributes this breakdown to the transition density becoming markedly non-Gaussian in nonlinear systems. Here, we provide a complementary dynamical perspective, and **show that the curvature of the underlying flow field and its evolution along trajectories** determines the bias introduced during inference in these settings. To mitigate this effect, we introduce a method that accounts for the flow field curvature by approximating the geodesic curves between consecutive observations computed on the Riemannian manifold induced by an estimate of the system's invariant density from the measurements. In experiments on nonlinear non-conservative Langevin systems, this geometric consideration substantially improves drift recovery at large sampling intervals.

## 1 INTRODUCTION

Stochastic differential equations (SDEs) are ubiquitous in science since they offer a versatile framework for describing processes with multiple degrees of freedom, like allele frequencies in a gene pool (Simpson et al., 2004), pollen motion in a liquid medium (Einstein, 1905), population dynamics (Silva-Dias & López-Castillo, 2018; Fisher & Mehta, 2014), chemical reactions (Li, 2020), cell growth (Alonso et al., 2014), and neural activity (Laing & Lord, 2009). A standard approach for inference of such systems is to approximate the drift function from observations by regressing state increments against the system state (Ruttor et al., 2013; Friedrich & Peinke, 1997; Ragwitz & Kantz, 2001). These approaches are essentially based on the Euler–Maruyama discretization, and work remarkably well when observations are densely sampled, i.e., when the inter-observation interval $\tau$ is small relative to the timescales of the underlying dynamics.

However, drift estimation degrades as $\tau$ grows, even when total observation time and noise level are fixed (Lade, 2009; Batz et al., 2018) (Fig. 1). A common probabilistic account attributes this breakdown to the transition density becoming markedly non-Gaussian for nonlinear systems at large $\tau$. Here we develop a complementary dynamical–geometric explanation that identifies the main source of bias as $\tau$ grows and suggests a natural correction.

The main intuition is the following: Euler–Maruyama-based inference implicitly treats the unobserved path between two samples as a straight segment in state space. In a nonlinear

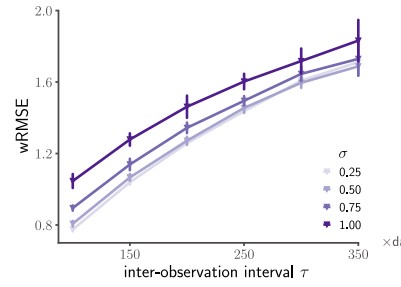

**Figure 1**
**Performance scaling with sampling interval $\tau$.**

---

[*]Preliminary versions of this work appeared in the workshop papers Maoutsa (2022; 2023), where the inference method was first introduced. The present paper contributes a theoretical analysis that further motivates that method. A full-length exposition is provided in Maoutsa (2025) with the precent theoretical analysis added in the supplement of the latest update.

system, trajectories bend, and their curvature changes as the state evolves. We show that the leading-order drift bias is controlled by the geometric quantity

$$\nabla(J_f\mathbf{f}) \cdot \mathbf{f}, \tag{1}$$

which measures how the flow acceleration (and thus the curvature) evolves along trajectories. Together with diffusion-weighted corrections, this induces a systematic $\mathcal{O}(\tau^2)$ bias that becomes dominant under sparse sampling.

**Contributions.** **(i)** We identify the Itô–Taylor remainder term driving the leading drift bias at large $\tau$, controlled by $\nabla(J_f f) \cdot f$ (plus diffusion corrections). **(ii)** We interpret this term geometrically as the correction to the straight-segment approximation implicit in Euler–Maruyama drift regression. **(iii)** We use this insight to motivate a curvature-aware inference method: we construct geodesic reference paths on a Riemannian manifold induced by an estimate of the invariant density, resulting in curvature-correcting constraints that improve drift inference for large sampling intervals.

## 2 METHODOLOGY

**Setting.** We consider a $d$-dimensional stochastic system whose state evolves according to the Itô SDE

$$d\mathbf{X}_t = \mathbf{f}(\mathbf{X}_t)\,dt + \sigma\,d\mathbf{W}_t, \tag{2}$$

with unknown drift $\mathbf{f} : \mathbb{R}^d \to \mathbb{R}^d$ and known constant diffusion $\sigma \in \mathbb{R}^{d \times d}$. We observe $\mathcal{O}_k \doteq \mathbf{X}_{t_k}$ at times $t_k = k\tau$, and want to infer $\mathbf{f}$ from $\{\mathcal{O}_k\}_{k=1}^K$.

**Conventional inference methods and their limitations.** A common drift estimator approximates the local drift by $\widehat{\mathbf{f}}(\mathbf{X}_{t_k}) \approx \frac{\mathbf{X}_{t_{k+1}} - \mathbf{X}_{t_k}}{\tau}$, which corresponds to the Euler–Maruyama (EuM) discretization scheme $\mathbf{X}_{t+\tau} \mid \mathbf{X}_t \approx \mathcal{N}(\mathbf{X}_t + \mathbf{f}(\mathbf{X}_t)\tau,\ \sigma\sigma^\top\tau)$ (Ruttor et al., 2013). While accurate for small $\tau$, EuM-based inference becomes systematically biased under sparse sampling; our goal here is to characterize the leading source of this bias (see Fig 4).

**Itô–Taylor remainder.** Writing Eq. 2 in integral form and expanding the drift over $[t_0, t_0 + \tau]$ results in

$$\mathbf{X}_{t_0+\tau} = \mathbf{X}_{t_0} + \mathbf{f}(\mathbf{X}_{t_0})\,\tau + \sigma(\mathbf{W}_{t_0+\tau} - \mathbf{W}_{t_0}) + R_1, \tag{3}$$

where $R_1$ comprises higher-order terms (see derivation in Appendix B). Euler–Maruyama effectively sets $R_1 \approx 0$, and thereby the sparse-sampling failure arises precisely when this remainder is no longer negligible.

## 3 RESULTS

**Leading-order bias from flow-curvature evolution.** The EuM approximation neglects the Itô–Taylor remainder $R_1$ in Eq. 3. A detailed expansion (Appendix B) shows that the dominant deterministic contribution can be written as a double integral of the backward generator acting on $\mathbf{f}$, and results in the following leading term

$$\frac{1}{\tau}R_{1,a}^1 \approx \frac{\tau^2}{2}\left[\nabla(J_f\mathbf{f}) \cdot \mathbf{f} + \nabla\left(\tfrac{1}{2}\Delta_D\mathbf{f}\right) \cdot \mathbf{f}\right], \tag{4}$$

where $J_f$ is the Jacobian of $\mathbf{f}$ and $\Delta_D$ is the diffusion-weighted Laplacian. The first term, $\nabla(J_f\mathbf{f})\cdot\mathbf{f}$, quantifies how the flow acceleration $\ddot{\mathbf{x}} = J_f\mathbf{f}$ evolves along trajectories of $\dot{\mathbf{x}} = \mathbf{f}(\mathbf{x})$, i.e. how trajectory curvature changes along the flow. Eq 4 therefore implies a **systematic drift bias** that scales as $\mathcal{O}(\tau^2)$ under sparse sampling, vanishes for linear drifts, and becomes large in regions where curvature varies very fast.

**Curvature-aware correction via empirical manifold geodesics.** The bias in Eq.4 arises from treating the latent path between $\mathcal{O}_k$ and $\mathcal{O}_{k+1}$ as approximately straight. We can mitigate this by constructing **geodesic reference paths** on a Riemannian manifold induced by the empirical invariant density. From observations $\{\mathcal{O}_k\}$ we estimate a (diagonal) local metric $H(x)$

$$H_{dd}(\mathbf{x}) = \left(\sum_{k=1}^K w_k(\mathbf{x})\big(\mathcal{O}_k^{(d)} - x^{(d)}\big)^2 + \epsilon\right)^{-1}, \qquad w_k(\mathbf{x}) = \exp\left(-\frac{\|\mathcal{O}_k - \mathbf{x}\|^2}{2\,\sigma_M^2}\right), \tag{5}$$

and compute the geodesics $\gamma_k^\star$ connecting consecutive observations by minimizing

$$\gamma_k^\star = \arg\min_{\gamma:\,\gamma(0)=\mathcal{O}_k,\,\gamma(1)=\mathcal{O}_{k+1}} \int_0^1 \|\dot{\gamma}(t')\|_{H(\gamma(t'))}^2\,dt'. \tag{6}$$

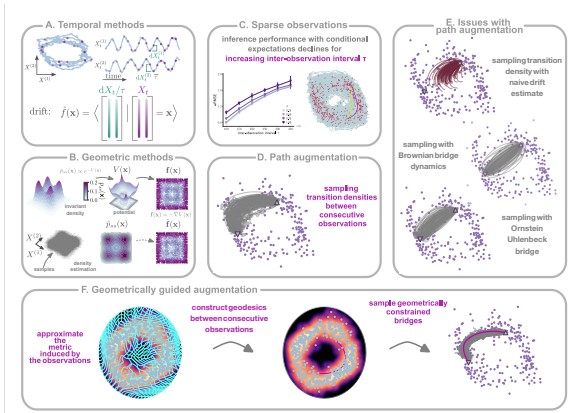

**Figure 2**
**Temporal and geometric perspectives for discovering stochastic dynamics and proposed inference with geometrically guided augmentation.**
(**A.**) Temporal methods consider the time-ordering of the observations (*purple dots*) and approximate the unknown drift function $\mathbf{f}(\mathbf{x})$ by the empirically measured conditional expectation of state increments given the system state, $\hat{f}(\mathbf{x}) = \langle \frac{d\mathbf{X}_t}{s} | \mathbf{X}_t = \mathbf{x} \rangle$. This amounts to solving a regression problem with measured states $\mathbf{X}_t$ and rescaled state increments $d\mathbf{X}_t/\tau$. (**B.**) Geometric approaches assume that the drift $\mathbf{f}(\mathbf{x})$ is the gradient of a potential $V(\mathbf{x})$ and thus are restricted only to conservative systems. (**C.**) Accuracy of predictions of temporal methods quantified by the weighted root mean square error (wRMSE) between predicted $\hat{\mathbf{f}(\mathbf{x})}$ and ground truth drift $\mathbf{f}(\mathbf{x})$ deteriorates with increasing inter-observation interval $\tau$. Geometrically this missestimation occurs because the Euclidean distance employed for computing the state increments does account for the curvature of the latent continuous path between consecutive observations. (**D.**) Path augmentation methods interleave state estimation between successive observations with inference in an iterative scheme, creating continuous trajectories for the unobserved parts of the sample path by sampling diffusion bridges for each inter-observation interval $k$ starting from $\mathcal{O}_k$ and ending at $\mathcal{O}_{k+1}$. (**E.**) Due to the computationally challenging nature of the problem, commonly used path augmentation methods employ Brownian or Ornstein-Uhlenbeck bridge dynamics, that fail to appropriately match the underlying unobserved for large $\tau$. (**F.**) Geometrically guided augmentation approximates first the metric induced by the invariant density, constructs geodesics connecting consecutive observations, and samples geometrically constrained diffusion bridges.

We then perform path augmentation by sampling bridges constrained to remain near $\gamma_k^\star$ (implemented as a stochastic control penalty), and estimate the drift $\mathbf{f}$ from the augmented paths. This enforces curvature-consistent interpolation between samples and reduces the $\mathcal{O}(\tau^2)$ bias.

## 4  EMPIRICAL VALIDATION

We validate our approach on Van der Pol (Fig 3) and Repressilator dynamics (Fig. 5) and report the weighted RMSE of the identified drift (weights proportional to empirical density). Consistent with Eq.4, baseline GP drift regression shows a pronounced superlinear degradation with $\tau$ (Van der Pol with larger $\mu$). Geodesic-guided augmentation substantially improves drift inference at large $\tau$ (where baseline estimates become substantially incorrect).

## 5  DISCUSSION

We have provided a geometric characterization of why standard drift inference fails under sparse sampling. We attributed the failure on the fact that conventional inference methods derived from the EuM approximation assume trajectories that are locally straight between observations, neglecting the curvature of the flow field. The additional term that becomes important for large intervals quantifies how trajectory curvature changes along the flow. Our analysis identifies when this mat-

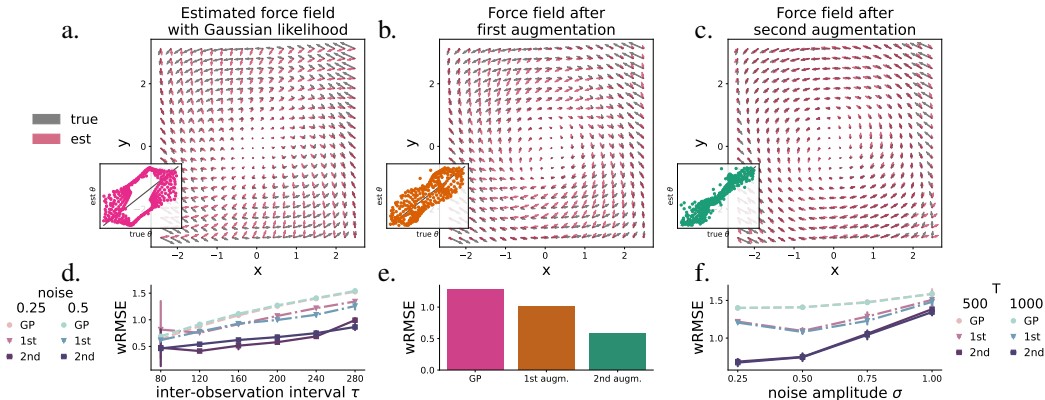

**Figure 3**
**Geometry-aware path augmentation improves drift inference after two iterations.** Estimated (*red*) vs. true (*grey*) force field with **a.)** Gaussian likelihood, **b.)** after one, and **c.)** after two augmentations. (**Insets**) True vs. estimated angles at grid points. **d.)** Weighted (by observation density) root mean square error (wRMSE) vs. inter-observation interval $\tau$ for different noise levels $\sigma = \{0.25, 0.5\}$ for drift estimated with a Gaussian likelihood (*gaus*-circles), after first augmentation (*1st*-triangles), and after second augmentation (*2nd*-squares) for $T = 500$ (time units). **e.)** wRMSE across iterations for the presented example. **f.)** wRMSE vs. noise amplitude $\sigma$ for different trajectory durations $T = \{500, 1000\}$ (time units) for inter-observation interval $\tau = 240$ ($dt$). Markers in **d.)** and **f.)** indicate augmentation steps. Error bars: one standard deviation over five independent runs.

ters: accuracy requires $\tau \ll \tau_{\mathrm{curv}}$, where $\tau_{\mathrm{curv}}$ is the timescale on which flow curvature varies significantly.

This geometric perspective suggests how to fix the problem: geodesics on the observation manifold provide natural reference paths that approximate the curvature of the learned geometry. Using these reference paths as constraints for path augmentation partially corrects the bias without requiring parametric assumptions about the drift function. Our experiments confirm that our approach works. Geodesic-guided inference recovers accurate drift estimates even at $\tau = 240dt$, where naive methods fail.

This connects to broader themes in geometric statistics and manifold learning. The manifold hypothesis (that high-dimensional data concentrates near low-dimensional structures) is typically invoked for dimensionality reduction or representation learning. Here, we show it also matters for dynamical inference. The geometry of the empirical manifold contains information about trajectory curvature that is essential to be accounted for when observations are sparse. Methods that ignore this geometry, treating state space as flat Euclidean space, inevitably introduce systematic bias.

There are natural extensions to explore. First, here we focus on constant diffusion. State-dependent diffusion and the metric induced from such a system needs to be additionally accounted for. Second, the stochastic component $R_{1,b}$ contributes variance but not bias, characterizing its scaling and interaction with the deterministic curvature term could help with the quantification of uncertainty.

## 6 CONCLUSION

We analysed the breakdown of common SDE inference methods under sparse sampling from a geometric perspective. We showed that it reflects the assumption of straight trajectories between consecutive observations, while the true underlying flow field might be curved. The dominant error is controlled by $\nabla(J_f \mathbf{f}) \cdot \mathbf{f}$, which quantifies the curvature evolution and results in an $\mathcal{O}(\tau^2)$ bias scaling as $\tau^2/\tau_{\mathrm{curv}}^2$. Geodesics on the Riemannian manifold induced by the invariant density provide curvature-approximating reference paths that account for this bias. Performing path augmentation by sampling sample paths conditioned to remain in the vicinity of those geodesic curves improves inference especially in the large inter-observation limit.

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

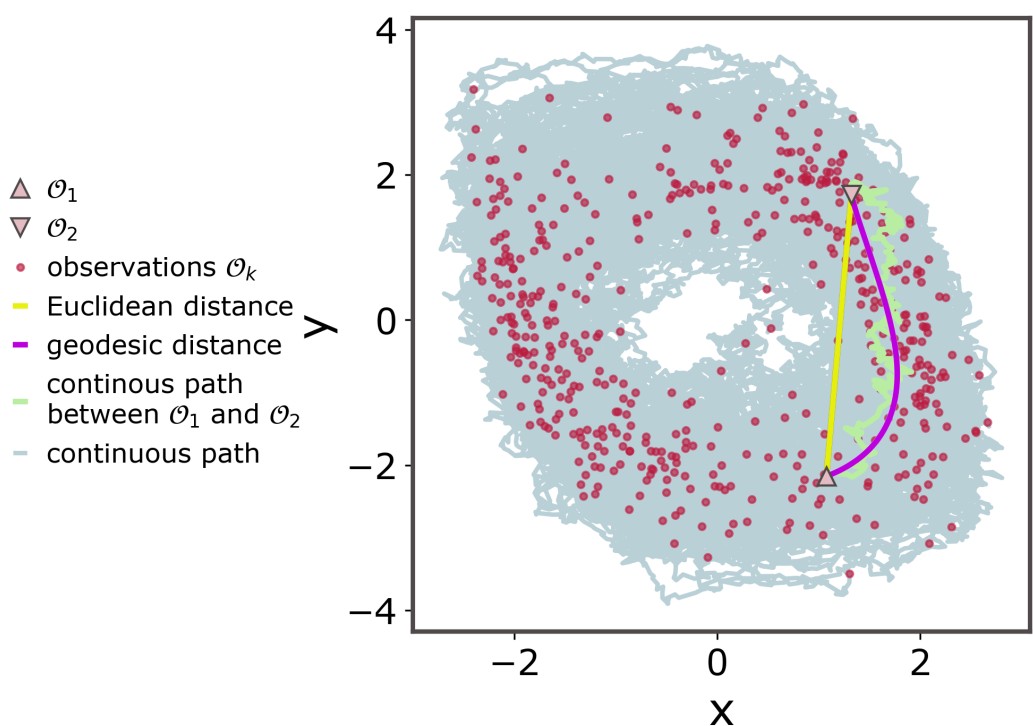

**Figure 4**

**Assumed path between consecutive observations (pink triangles) according to conventional drift inferences approaches (yellow line), true underlying path (neon green line), and geodesic curve (magenta line) employed by our method as constraint for the unobserved path.**

## A    ADDITIONAL FIGURES

### A.1    INFERENCE FOR A 3-DIMENSIONAL SYSTEM

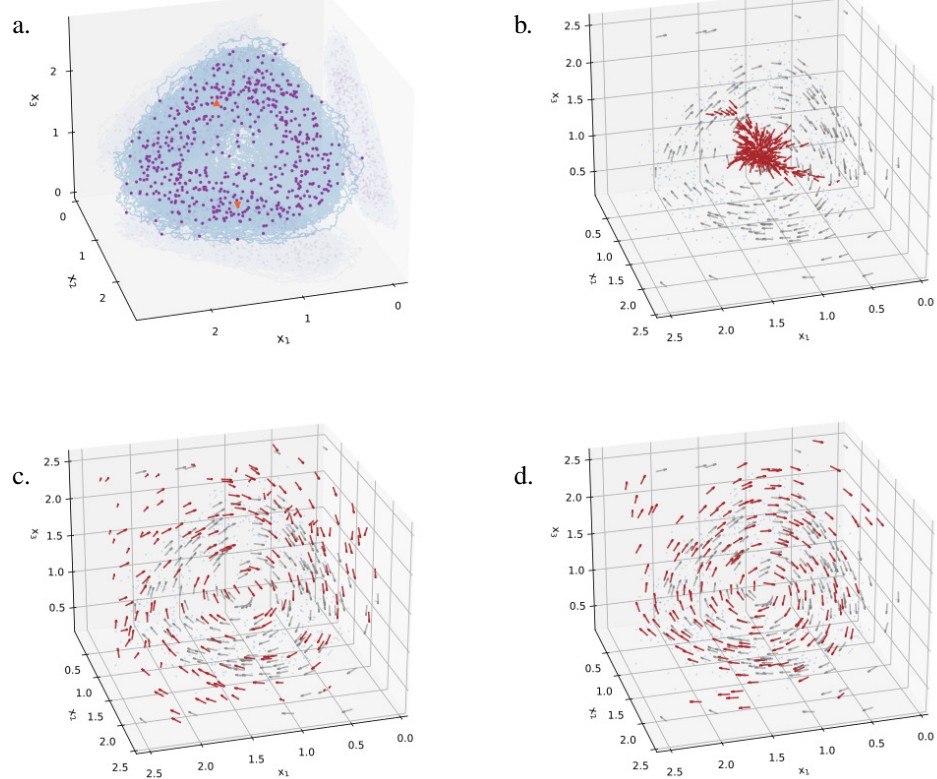

**Figure 5**
**Inference for a three dimensional system following Repressilator dynamics.** **(a.)**Empirical manifold of the Repressilator system. Purple dots indicate the observations and blue line denotes the continuous unobserved trajectory. Orange triangles indicate two consecutive observations. **(b.)** Ground truth (*grey*) and estimated (*maroon*) flow field with Gaussian process regression without augmentation **(c.-d.)** Ground truth (*grey*) and estimated (*maroon*) flow field with the proposed framework after (c.) one and (d.) two augmentations.

## B    INFERENCE BASED ON EULER-MARUYAMA DISCRETISATION DOES NOT ACCOUNT FOR THE CURVATURE OF THE TRAJECTORIES IN THE STATE SPACE

To be more precise, a general SDE of the form

$$d\mathbf{X}_t = \mathbf{f}(\mathbf{X}_t, t)dt + \boldsymbol{\sigma}(\mathbf{X}_t, t)d\mathbf{W}_t. \tag{7}$$

is a shorthand for the integral equation

$$\mathbf{X}_t = \mathbf{X}_{t_0} + \int_{t_0}^{t} \mathbf{f}(\mathbf{X}_s, s)\,ds + \int_{t_0}^{t} \boldsymbol{\sigma}(\mathbf{X}_s, s)\,d\mathbf{W}_s, \tag{8}$$

where as previously in this manuscript, we consider the stochastic integrals in the **Itô sense**. (Here we start from a more general formulation of the stochastic equation with both diffusion and drift terms being state- and time-dependent to highlight that also for more general SDEs our geometric argument is valid.)

Applying the Itô formula on each integrand, and integrating from $t_0$ to $t$, we obtain the Itô-Taylor expansion of Eq. 7

$$
\begin{aligned}
\mathbf{f}(\mathbf{X}_t, t) =&\, \mathbf{f}(\mathbf{X}_{t_0}, t_0) + \int_{t_0}^{t} \frac{\partial \mathbf{f}(\mathbf{X}_s, s)}{\partial s}\, \mathrm{d}s + \int_{t_0}^{t} \sum_u \frac{\partial \mathbf{f}(\mathbf{X}_s, s)}{\partial X^{(u)}} f_u(\mathbf{X}_s, s)\, \mathrm{d}s \\
&+ \int_{t_0}^{t} \sum_u \frac{\partial \mathbf{f}(\mathbf{X}_s, s)}{\partial X^{(u)}} [\boldsymbol{\sigma}(\mathbf{X}_s, s)\, \mathrm{d}\mathbf{W}_s]_u + \int_{t_0}^{t} \frac{1}{2} \sum_{u,v} \frac{\partial^2 \mathbf{f}(\mathbf{X}_s, s)}{\partial X^{(u)} \partial X^{(v)}} [\boldsymbol{\sigma}(\mathbf{X}_s, s)\, \boldsymbol{\sigma}^\top(\mathbf{X}_s, s)]_{uv}\, \mathrm{d}s \\
=&\, \mathbf{f}(\mathbf{X}_{t_0}, t_0) + \int_{t_0}^{t} \mathcal{L}_s^\dagger \mathbf{f}(\mathbf{X}_s, s)\, \mathrm{d}s + \sum_\nu \int_{t_0}^{t} \mathcal{L}_{W,\nu} \mathbf{f}(\mathbf{X}_s, s)\, \mathrm{d}W_s^{(\nu)},
\end{aligned}
\tag{9}
$$

and

$$
\begin{aligned}
\boldsymbol{\sigma}(\mathbf{X}_t, t) =&\, \boldsymbol{\sigma}(\mathbf{X}_{t_0}, t_0) + \int_{t_0}^{t} \frac{\partial \boldsymbol{\sigma}(\mathbf{X}_s, s)}{\partial s}\, \mathrm{d}s + \int_{t_0}^{t} \sum_u \frac{\partial \boldsymbol{\sigma}(\mathbf{X}_s, s)}{\partial X^{(u)}} f_u(\mathbf{X}_s, s)\, \mathrm{d}s \\
&+ \int_{t_0}^{t} \sum_u \frac{\partial \boldsymbol{\sigma}(\mathbf{X}_s, s)}{\partial X^{(u)}} [\boldsymbol{\sigma}(\mathbf{X}_s, s)\, \mathrm{d}\mathbf{W}_s]_u + \int_{t_0}^{t} \frac{1}{2} \sum_{u,v} \frac{\partial^2 \boldsymbol{\sigma}(\mathbf{X}_s, s)}{\partial X^{(u)} \partial X^{(v)}} [\boldsymbol{\sigma}(\mathbf{X}_s, s)\, \boldsymbol{\sigma}^\top(\mathbf{X}_s, s)]_{uv}\, \mathrm{d}s \\
=&\, \boldsymbol{\sigma}(\mathbf{X}_{t_0}, t_0) + \int_{t_0}^{t} \mathcal{L}_s^\dagger \boldsymbol{\sigma}(\mathbf{X}_s, s)\, \mathrm{d}s + \sum_\nu \int_{t_0}^{t} \mathcal{L}_{W,\nu} \boldsymbol{\sigma}(\mathbf{X}_s, s)\, \mathrm{d}W_s^{(\nu)},
\end{aligned}
\tag{10}
$$

where we have used the fact that the product of stochastic differentials due to the Ito isometry and multiplication rules equals the noise covariance times the time step

$$
dX_t^{(u)} dX_t^{(v)} = \left[\boldsymbol{\sigma}\boldsymbol{\sigma}^\top\right]_{uv} dt,
$$

where

$$
dX_s^{(u)} = f_u\, ds + \sum_{j=1}^{m} \sigma_{uj}\, dW_s^{(j)},
$$

while the superscripts/subscripts $u, v$ indicate dimensional components.

In the above equations, we have introduced the operators acting on test-functions $\mathbf{h} : \mathcal{R}^D \to \mathcal{R}^D$

$$
\mathcal{L}_t^\dagger \mathbf{h} = \frac{\partial \mathbf{h}}{\partial t} + \sum_u \frac{\partial \mathbf{h}}{\partial X^{(u)}} f_u + \frac{1}{2} \sum_{u,v} \frac{\partial^2 \mathbf{h}}{\partial X^{(u)} \partial X^{(v)}} \left[\boldsymbol{\sigma}(\mathbf{X}_s, s)\, \boldsymbol{\sigma}^\top(\mathbf{X}_s, s)\right]_{uv}
\tag{11}
$$

and

$$
\mathcal{L}_{W,v} \mathbf{h} = \sum_u \frac{\partial \mathbf{h}}{\partial X^{(u)}}\, \boldsymbol{\sigma}_{uv}, \qquad \text{for } v = 1, \ldots, n.
\tag{12}
$$

With these expressions, the original integral equation for $\mathbf{X}_t$ can be written as

$$
\mathbf{X}_t = \mathbf{X}_{t_0} + \mathbf{f}(\mathbf{X}_{t_0}, t_0)(t - t_0) + \boldsymbol{\sigma}(\mathbf{X}_{t_0}, t_0)\left(\mathbf{W}_t - \mathbf{W}_{t_0}\right) +
\tag{13}
$$

$$
{\color{orange} R_1 =}\ \begin{cases} + \displaystyle\int_{t_0}^{t} \int_{t_0}^{s} \mathcal{L}_u^\dagger \mathbf{f}(\mathbf{X}_u, u)\, \mathrm{d}u\, \mathrm{d}s + \sum_\nu \int_{t_0}^{t} \int_{t_0}^{s} \mathcal{L}_{W,\nu} \mathbf{f}(\mathbf{X}_u, u)\, \mathrm{d}W_u^{(\nu)}\, \mathrm{d}s \\[2ex] + \displaystyle\int_{t_0}^{t} \int_{t_0}^{s} \mathcal{L}_u^\dagger \boldsymbol{\sigma}(\mathbf{X}_u, u)\, \mathrm{d}u\, \mathrm{d}\mathbf{W}_s + \sum_\nu \int_{t_0}^{t} \int_{t_0}^{s} \mathcal{L}_{W,\nu} \boldsymbol{\sigma}(\mathbf{X}_u, u)\, \mathrm{d}W_u^{(\nu)}\, \mathrm{d}\mathbf{W}_s. \end{cases}
$$

In the last equation, dropping the terms in the remainder $R_1$ results in the Euler–Maruyama integration scheme (Jentzen & Kloeden, 2011). Introducing the discrete time and noise increments

$$
\Delta t_n = t_{n+1} - t_n = \int_{t_n}^{t_{n+1}} \mathrm{d}s, \quad \Delta \mathbf{W}_n = \mathbf{W}_{t_{n+1}} - \mathbf{W}_{t_n} = \int_{t_n}^{t_{n+1}} \mathrm{d}\mathbf{W}_s,
\tag{14}
$$

we result in the discrete time equation commonly used for numerical integration of SDEs

$$
\mathbf{X}_{n+1} = \mathbf{X}_n + \mathbf{f}(\mathbf{X}_n, t_n)\, \Delta t_n + \boldsymbol{\sigma}\, \Delta \mathbf{W}_n.
\tag{15}
$$

This is also the starting point of most inference methods that employ the regression scheme mentioned above by approximating the drift as

$$\hat{\mathbf{f}}(\mathbf{X}_n, t_n) \approx \frac{\mathbf{X}_{n+1} - \mathbf{X}_n}{\Delta t} \sim \mathcal{N}\left(\hat{\mathbf{f}}(\mathbf{X}_n, t_n), \frac{\boldsymbol{\sigma}\,\boldsymbol{\sigma}^\top}{\Delta t}\right). \tag{16}$$

This discretisation is a zero-order approximation of the true dynamics, and assumes that $\mathbf{f}(\cdot)$ remains constant throughout the interval $\Delta t$, i.e. throughout the inter-observation interval $\tau$ in the inference setting. However as $\tau$ increases, higher-order terms in the remainder $R_1$ of the Itô-Taylor expansion become significant, since the assumption that the drift is approximately constant over $\tau$ does not hold.

We can glean onto the terms that become important once the inter-observation interval becomes large, by applying the Itô formula onto each one of the integrands in $R_1$ separately **for the specific setting we consider in this manuscript**, i.e. that of time-independent drift function $\mathbf{f}(\mathbf{x})$ and constant diffusion matrix $\boldsymbol{\sigma}$. In the following, we demonstrate that the leading-order error of this approximation is governed by the intrinsic geometry of the drift vector field. This provides further insight and a geometric explanation for the deterioration of inference methods for increasing inter-observation interval $\tau$.

In short we show that, inference methods based on the Euler-Maruyama discretisation-based inference effectively assume that the vector field between consecutive observations $\mathbf{X}_n$ and $\mathbf{X}_{n+1}$ does not change. Our analysis shows this is equivalent to assuming trajectories are straight lines ($\mathbf{J}_f \mathbf{f} \parallel \mathbf{f}$) and the Itô correction is constant. In reality, trajectories curve ($\mathbf{J}_f \mathbf{f}$ has also a perpendicular component), and this curvature itself changes along the vector field. The Euler-Maruyama discretisation-based inference scheme systematically misses these higher-order geometric features, leading to biased drift estimates.

## B.1 First remainder term $R_{1,a}$

We denote the first term of the reminder by $R_{1,a}$

$$R_{1,a} = \int_{t_0}^{t} \int_{t_0}^{s} \mathcal{L}_u^\dagger \mathbf{f}(\mathbf{X}_u)\,\mathrm{d}u\,\mathrm{d}s. \tag{17}$$

Applying Itô's formula to the integrand $\mathcal{L}_t^\dagger \mathbf{f}(\mathbf{X}_u, u)$, we get

$$\mathrm{d}\mathcal{L}_u^\dagger \mathbf{f}(\mathbf{X}_u) = \frac{\partial}{\partial u}\mathcal{L}_u^\dagger \mathbf{f}(\mathbf{X}_u)\,\mathrm{d}u + \sum_{j=1}^{d} \frac{\partial \mathcal{L}_u^\dagger \mathbf{f}}{\partial X^{(j)}}(\mathbf{X}_u)\,\mathrm{d}X_u^{(j)} + \frac{1}{2}\sum_{j,k=1}^{d} \frac{\partial^2 \mathcal{L}_u^\dagger \mathbf{f}}{\partial X^{(j)}\partial X^{(k)}}(\mathbf{X}_u)\,\left[\boldsymbol{\sigma}\boldsymbol{\sigma}^\top\right]_{jk}\,\mathrm{d}u. \tag{18}$$

Plugging in the original equation $\mathrm{d}X_u^{(j)} = f_j\,\mathrm{d}u + \sum_{\nu=1}^{m} \sigma_{j\nu}\,\mathrm{d}W_u^{(\nu)}$, and integrating over the time from $t_0$ to $u$

$$\mathcal{L}_u^\dagger \mathbf{f}(\mathbf{X}_u) = \mathcal{L}_{t_0}^\dagger \mathbf{f}(\mathbf{X}_{t_0}) + \int_{t_0}^{u}\left(\frac{\partial}{\partial w}(\mathcal{L}_w^\dagger \mathbf{f}(\mathbf{X}_w)) + \sum_{j} \frac{\partial(\mathcal{L}_w^\dagger \mathbf{f})}{\partial X^{(j)}}f_j + \frac{1}{2}\sum_{j,k} \frac{\partial^2(\mathcal{L}_w^\dagger \mathbf{f})}{\partial X^{(j)}\partial X^{(k)}}[\boldsymbol{\sigma}\boldsymbol{\sigma}^\top]_{jk}\right)\,\mathrm{d}w$$
$$+ \int_{t_0}^{u}\sum_{j} \frac{\partial(\mathcal{L}_w^\dagger \mathbf{f})}{\partial X^{(j)}}[\boldsymbol{\sigma}\mathrm{d}\mathbf{W}_w]_j. \tag{19}$$

Applying Fubini's theorem in the original double integral, we change the order of integration

$$\int_{t_0}^{t} \int_{t_0}^{s} \phi(u)\,\mathrm{d}u\,\mathrm{d}s = \int_{t_0}^{t} (t-u)\,\phi(u)\,\mathrm{d}u, \tag{20}$$

and applying it twice we obtain

$$R_{1,a} = \int_{t_0}^{t} \int_{t_0}^{s} \mathcal{L}_u^{\dagger} \mathbf{f}(\mathbf{X}_u) \, \mathrm{d}u \, \mathrm{d}s = \int_{t_0}^{t} \frac{(t-u)^2}{2} \left[ \underbrace{\sum_j \frac{\partial \mathcal{L}_u^{\dagger} \mathbf{f}}{\partial X^{(j)}} f_j}_{R_{1,a}^1} + \underbrace{\frac{1}{2} \sum_{j,k} \frac{\partial^2 \mathcal{L}_u^{\dagger} \mathbf{f}}{\partial X^{(j)} \partial X^{(k)}} [\boldsymbol{\sigma}\boldsymbol{\sigma}^{\top}]_{jk}}_{R_{1,a}^2} \right] \mathrm{d}u$$

$$+ \underbrace{\int_{t_0}^{t} \frac{(t-u)^2}{2} \sum_j \frac{\partial \mathcal{L}_u^{\dagger} \mathbf{f}}{\partial X^{(j)}} [\boldsymbol{\sigma} \, \mathrm{d}\mathbf{W}_u]_j}_{R_{1,a}^3} + \frac{\tau^2}{2} \mathcal{L}_t^{\dagger} \mathbf{f}(\mathbf{X}_{t_0}). \qquad (21)$$

In the previous equation we have dropped the term $\frac{\partial}{\partial w}\left(\mathcal{L}_w^{\dagger} \mathbf{f}(\mathbf{X}_w)\right)$ that is equal to zero and that would require the drift $\mathbf{f}$ to be time-dependent to be non-negligible.

**First component $R_{1,a}^1$ of remainder term $R_{1,a}$: Flow curvature term.**   The Backward Kolmogorov generator applied to a vector field $\mathbf{f}$ can be written as

$$\mathcal{L}^{\dagger} \mathbf{f} = \mathbf{J}_f \mathbf{f} + \frac{1}{2} \Delta_D \mathbf{f}. \qquad (22)$$

In Eq. 22, $\mathbf{J}_f \doteq \nabla \mathbf{f}$ denotes the Jacobian of $\mathbf{f}$, $\mathbf{D} \doteq \boldsymbol{\sigma}\boldsymbol{\sigma}^{\top}$ the noise covariance, and $\Delta_{\mathbf{D}} \doteq \sum_{j,k} \mathbf{D}_{jk} \partial^2_{X^{(j)} X^{(k)}}$ is the noise-covariance weighted Laplacian operator. Thus each component of $\mathcal{L}^{\dagger}\mathbf{f}$ comprises the directional derivative of the drift $\mathbf{J}_f \mathbf{f}$ plus an anisotropic/noise-covariance weighted Laplacian of $\mathbf{f}$, which in component-wise form is expressed as

$$\left[\mathcal{L}^{\dagger}\mathbf{f}\right]_i = \sum_k \frac{\partial f_i}{\partial X^{(k)}} f_k + \frac{1}{2} \sum_{k,\ell} \mathbf{D}_{k\ell} \frac{\partial^2 f_i}{\partial X^{(k)} \partial X^{(\ell)}}. \qquad (23)$$

Differentiating wrt to $X^{(j)}$ yields

$$\frac{\partial}{\partial X^{(j)}} \left[\mathcal{L}^{\dagger}\mathbf{f}\right]_i = \sum_k \frac{\partial^2 f_i}{\partial X^{(j)} \partial X^{(k)}} f_k + \sum_k \frac{\partial f_i}{\partial X^{(k)}} \frac{\partial f_k}{\partial X^{(j)}} + \frac{1}{2} \sum_{k,\ell} \mathbf{D}_{k\ell} \frac{\partial^3 f_i}{\partial X^{(j)} \partial X^{(k)} \partial X^{(\ell)}},$$
$$(24)$$

and thus we rewrite the $i$-th component of the term $R_{1,a}^1$ as

$$\left[R_{1,a}^1\right]_i = \int_{t_0}^{t} \frac{(t-u)^2}{2} \left[ \sum_{j,k} \frac{\partial^2 f_i}{\partial X^{(j)} \partial X^{(k)}} f_k f_j + \sum_{j,k} \frac{\partial f_i}{\partial X^{(k)}} \frac{\partial f_k}{\partial X^{(j)}} f_j + \frac{1}{2} \sum_{j,k,\ell} \mathbf{D}_{k\ell} \frac{\partial^3 f_i}{\partial X^{(j)} \partial X^{(k)} \partial X^{(\ell)}} f_j \right]_i \mathrm{d}u.$$
$$(25)$$

The third-order state-derivative in the last summand of Eq. 25, indicates that this last term is inactive for linear or quadratic drift functions $\mathbf{f}$.

We re-write again this part of the remainder in a more compact vector notation in terms of the directional derivative of $(\mathbf{J}_f \mathbf{f})$ and $\frac{1}{2} \Delta_D \mathbf{f}$ along the vector field as

$$R_{1,a}^1 = \int_{t_0}^{t} \frac{(t-u)^2}{2} \Big[ \underbrace{\nabla(\mathbf{J}_f \mathbf{f}) \cdot \mathbf{f}}_{\text{flow curvature}} + \underbrace{\nabla(\tfrac{1}{2} \Delta_D \mathbf{f}) \cdot \mathbf{f}}_{\text{diffusive term along the flow}} \Big] \mathrm{d}u. \qquad (26)$$

This part of the remainder captures two geometric effects that standard inference methods neglect: the **intrinsic curvature of deterministic flow trajectories in state space**, and the **systematic bias introduced by the spatial variation of both drift and diffusion** along these trajectories, when both drift and diffusion are assumed as constant between inter-observation intervals.

- To understand the **first term**, $\nabla(\mathbf{J}_f \mathbf{f}) \cdot \mathbf{f}$, from a geometric perspective, let us consider a deterministic dynamical system with dynamics $\dot{\mathbf{x}}_t = \mathbf{f}(\mathbf{x}_t)$. A trajectory initiated from an

initial condition $\mathbf{x}_0$ traces a streamline in the state space $\mathcal{R}^d$. We express the acceleration of this trajectory in terms of the directional derivative

$$\ddot{\mathbf{x}}_t = \frac{\mathrm{d}}{\mathrm{d}t}\mathbf{f}(\mathbf{x}_t) = \mathbf{J}_f(\mathbf{x}_t) \cdot \mathbf{f}(\mathbf{x}_t) = \mathbf{J}_f \cdot \mathbf{f}. \tag{27}$$

The acceleration vector admits a natural orthogonal decomposition comprising a component parallel to the vector field $\mathbf{f}$ and an orthogonal component to $\mathbf{f}$

$$\mathbf{J}_f \cdot \mathbf{f} = P_\parallel(\mathbf{f})\,\mathbf{J}_f \cdot \mathbf{f} + P_\perp(\mathbf{f})\,\mathbf{J}_f \cdot \mathbf{f}. \tag{28}$$

Here $P_\parallel(\mathbf{f}(\mathbf{x})) = \frac{\mathbf{f}(\mathbf{x})\mathbf{f}^\top(\mathbf{x})}{\|\mathbf{f}(\mathbf{x})\|^2}$ and $P_\perp(\mathbf{f}(\mathbf{x})) = \mathbb{I} - P_\parallel(\mathbf{f}(\mathbf{x}))$ stand for the parallel and orthogonal projectors. The parallel component quantifies the rate of change of speed along the trajectory (tangential acceleration), whilst the perpendicular component defines the **curvature vector** $\kappa_{\text{flow}}(x)$ (Kühnel, 2002), which quantifies the bending of the trajectories

$$\boldsymbol{\kappa}_{\text{flow}}(\mathbf{x}) = \frac{P_\perp(\mathbf{f}(\mathbf{x}))\mathbf{J}_f(\mathbf{x})\mathbf{f}(\mathbf{x})}{\|\mathbf{f}(\mathbf{x})\|^2}. \tag{29}$$

When $\boldsymbol{\kappa}_{\text{flow}} = 0$, the trajectories are straight lines in the state space, while when $\|\boldsymbol{\kappa}_{\text{flow}}\| > 0$ they are curved.

The term $\nabla(\mathbf{J}_f\mathbf{f}) \cdot \mathbf{f}$ quantifies the **evolution of the trajectory curvature** [1] as the system moves along the flow field. From Eq. 25 we have for each dimensional component $i$ of this term

$$\begin{aligned}
[\nabla(\mathbf{J}_f\mathbf{f}) \cdot \mathbf{f}]_i &= \sum_{j,k} \frac{\partial^2 f_i}{\partial X^{(j)}\partial X^{(k)}} f_k f_j + \sum_{j,k} \frac{\partial f_i}{\partial X^{(k)}} \frac{\partial f_k}{\partial X^{(j)}} f_j \\
&= [\mathbf{f}^\top(\nabla^2 f_i)\mathbf{f}] + [\mathbf{J}_f^2\mathbf{f}]_i.
\end{aligned} \tag{30}$$

We observe that this term captures the effects of how both second-order spatial variation of the flow field (the Hessian $\nabla^2 f_i$) and the Jacobian of the acceleration ($\mathbf{J}_f^2\mathbf{f}$) influence the evolution of trajectories.

- In Eq. 30, the **first sub-term**, $\mathbf{f}^\top(\nabla^2 f_i)\mathbf{f}$, represents the **second directional derivative** (or quadratic variation) of $f_i$ along the flow direction $\mathbf{f}$. It measures the curvature or second-order spatial variation of the $i$-th component of $\mathbf{f}$ in the direction $\mathbf{f}$. In regions where the Hessian $\nabla^2\mathbf{f}$ is large (as is for the case of a highly nonlinear drift with curving or bending behaviour), this term becomes significant, and it vanishes for linear or constant drift $\mathbf{f}$. Neglecting this term corresponds to approximating the flow by its linearisation.
- The **second sub-term**, $\mathbf{J}_f^2\mathbf{f} = \mathbf{J}_f(\mathbf{J}_f\mathbf{f})$, of Eq. 30 represents the action of the Jacobian operator on the acceleration vector. Geometrically, it describes how the local linearised field acts on the acceleration as we move an infinitesimal step along the flow field, or in other words how the linear approximation changes when following the flow direction $\mathbf{f}$.

By temporal integration we have

$$R_{1,a}^1 = \int_{t_0}^t \frac{(t-u)^2}{2}\left(\nabla(\mathbf{J}_f\mathbf{f})\cdot\mathbf{f} + \nabla\left(\tfrac{1}{2}\Delta_D\mathbf{f}\right)\cdot\mathbf{f}\right)\mathrm{d}u \sim \frac{\tau^2}{2}\left(\nabla(\mathbf{J}_f\mathbf{f})\cdot\mathbf{f} + \nabla\left(\tfrac{1}{2}\Delta_D\mathbf{f}\right)\cdot\mathbf{f}\right), \tag{31}$$

indicating that the evolution of trajectory curvature introduces an $O(\tau^3)$ correction to the transition density.

Drift inference based on Euler–Maruyama–type discretisation ignores between others the term $R_{1,a}^1$ introducing thereby a mean bias at each point $\mathbf{x}$ in the state space,

$$\beta_{1,a}^1(\mathbf{x}) = \frac{1}{\tau} R_{1,a}^1 \approx \frac{\tau}{2}\left[\nabla(\mathbf{J}_f\mathbf{f})\cdot\mathbf{f} + \nabla\left(\tfrac{1}{2}\Delta_D\mathbf{f}\right)\cdot\mathbf{f}\right](\mathbf{x}). \tag{32}$$

---

[1]More precisely the directional derivative of the acceleration, $\mathbf{J}_f(\mathbf{x})\cdot\mathbf{f}$ along the flow direction, or the **rate at which the acceleration changes along the flow, or a measure of how the local curvature of $\mathbf{f}$ as a vector field influences trajectory evolution**.

This bias induces a mean error in drift estimate, when using Euler–Maruyama-based inference, leading to under- or over-estimation of the true drift at state $\mathbf{x}$. This error scales as $\mathcal{O}(\tau^2)$ with the interval $\tau$.

Let us now consider the temporal rate of change experienced by a particle travelling along the flow field. The instantaneous speed of the particle at location $\mathbf{x}$ is $\|\mathbf{f}(\mathbf{x})\|$. The quantity in the brackets in Eq. 32, $\nabla(\mathbf{J}_f\mathbf{f}) \cdot \mathbf{f} + \nabla\left(\frac{1}{2}\Delta_D\mathbf{f}\right) \cdot \mathbf{f}$, is a spatial derivative measuring how quickly the curvature and diffusion variation change as the particle moves in space. The rate of change of this variation per unit of time is expressed as

$$\frac{\left\|\nabla(\mathbf{J}_f\mathbf{f}) \cdot \mathbf{f} + \nabla\left(\frac{1}{2}\Delta_D\mathbf{f}\right) \cdot \mathbf{f}\right\|(\mathbf{x})}{\|\mathbf{f}(\mathbf{x})\|} \doteq \tau_{\mathrm{curv}}^{-2}(\mathbf{x}). \tag{33}$$

In the last equation we have introduced the time scale of change $\tau_{\mathrm{curv}}$ as the inverse of the rate of change, which captures the characteristic time it takes for the curvature/diffusion variation to change significantly along the particles trajectory. Then the relative magnitude error in the Euler-Maruyama-based drift estimate satisfies

$$\frac{\|\beta_{1,a}^1(\mathbf{x})\|}{\|\mathbf{f}(\mathbf{x})\|} = \frac{\tau^2}{6\,\tau_{\mathrm{curv}}^2(\mathbf{x})}, \tag{34}$$

implying that the estimate is reliable only when the inter-observation interval $\tau \ll \tau_{\mathrm{curv}}(\mathbf{x})$.

- The **second term** in Eq.26, $\nabla(\frac{1}{2}\Delta_D\mathbf{f}) \cdot \mathbf{f}$, accounts for the diffusion part of the backward generator acting on the vector field $\mathbf{f}$. The anisotropic Laplacian $\Delta_D\mathbf{f}$ quantifies the **diffusion–weighted second-order spatial variation of the vector field**

$$[\Delta_D\mathbf{f}]_i = \sum_{j,k} D_{jk}\frac{\partial^2 f_i}{\partial X^{(j)}\partial X^{(k)}} = \nabla\cdot(\mathbf{D}\,\nabla f_i). \tag{35}$$

The directional derivative quantifies how this term evolves along the flow field

$$\left[\nabla\left(\frac{1}{2}\Delta_D\mathbf{f}\right) \cdot \mathbf{f}\right]_i = \frac{1}{2}\sum_{j,k,\ell} D_{k\ell}\frac{\partial^3 f_i}{\partial X^{(j)}\partial X^{(k)}\partial X^{(\ell)}}f_j. \tag{36}$$

This term captures how the diffusion-weighted spatial variation of the flow field varies across the state space. As trajectories traverse regions of varying drift curvature, the effective Itô correction itself changes, introducing systematic bias in inference methods that assume that drift is piece-wise constant in-between observations.

**Second component $R_{1,a}^2$ of remainder term $R_{1,a}$.**   The second component of the remainder term $R_{1,a}$ reads

$$R_{1,a}^2 = \int_{t_0}^t (t-u)\frac{1}{2}\sum_{j,k}\frac{\partial^2\left(\mathcal{L}_u^\dagger\mathbf{f}\right)}{\partial X^{(j)}\partial X^{(k)}}\left[\boldsymbol{\sigma}\boldsymbol{\sigma}^\top\right]_{jk}\mathrm{d}u. \tag{37}$$

For the $i$-th dimensional component we have

$$\frac{\partial^2}{\partial X^{(h)}\partial X^{(j)}}\left[\mathcal{L}_u^\dagger f\right]_i = \sum_k \frac{\partial^3 f_i}{\partial X^{(h)}\partial X^{(j)}\partial X^{(k)}}f_k + \sum_k \frac{\partial^2 f_i}{\partial X^{(j)}\partial X^{(k)}}\frac{\partial f_k}{\partial X^{(h)}}$$
$$+ \sum_k \frac{\partial^2 f_i}{\partial X^{(h)}\partial X^{(k)}}\frac{\partial f_k}{\partial X^{(j)}} + \sum_k \frac{\partial f_i}{\partial X^{(k)}}\frac{\partial^2 f_k}{\partial X^{(h)}\partial X^{(j)}} \tag{38}$$
$$+ \frac{1}{2}\sum_{k,\ell}\mathbf{D}_{k\ell}\frac{\partial^4 f_i}{\partial X^{(h)}\partial X^{(j)}\partial X^{(k)}\partial X^{(\ell)}}.$$

For this remainder term, we have for each dimensional component $i$

$$\left[R_{1,a}^2\right]_i = \int_{t_0}^t (t-u)\frac{1}{2}\sum_{j,k}\mathbf{D}_{jk}\left[\frac{\partial^2}{\partial X^{(k)}\partial X^{(j)}}\left[\mathcal{L}_u^\dagger\mathbf{f}\right]_i\right]\mathrm{d}u\,. \tag{39}$$

Geometrically, $R_{1,a}^2$ captures the **diffusion-weighted second-order spatial variation** of the generator $\mathcal{L}_u^\dagger \mathbf{f}$ across the $\sqrt{\tau}$-sized ellipsoid set by $\mathbf{D}$, i.e. the anisotropic Laplacian $\Delta_D(\mathcal{L}_u^\dagger \mathbf{f})$, the diffusion-weighted second spatial variation of the drift along the flow. Dropping this term in inference amounts to assuming $\mathcal{L}_u^\dagger \mathbf{f}$ is locally flat and results in an $O(\tau)$ drift bias of size $\beta_{1,a}^2 \approx (\tau/4)\,\Delta_D(\mathcal{L}_u^\dagger \mathbf{f})$, underestimating anisotropy and the evolution of curvature of the flow field, so inferred flow-lines appear too straight.

**Third component $R_{1,a}^3$ of remainder term $R_{1,a}$.**

$$R_{1,a}^3 = \int_{t_0}^t (t-u) \sum_j \frac{\partial \mathcal{L}_u^\dagger \mathbf{f}}{\partial X^{(j)}} [\sigma\,\mathrm{d}\mathbf{W}_u]_j, \tag{40}$$

$$\left[R_{1,a}^3\right]_i = \int_{t_0}^t (t-u) \sum_{j,m} \frac{\partial}{\partial X^{(j)}} \left[\mathcal{L}_u^\dagger \mathbf{f}\right]_i \sigma_{jm}\,\mathrm{d}\mathbf{W}_u^{(m)}, \tag{41}$$

This is a martingale term capturing the stochastic coupling between diffusion and the spatial inhomogeneity of the generator. In inference, this term doesn't introduce bias, since $\langle R_{1,a}^3 \rangle = 0$. However, neglecting this term, ignores a second–order variance contribution with $\mathrm{Var}(R_{1,a}^3/\tau) = O(\tau)$.

B.2  Second remainder term $R_{1,b}$

We denote the second term of the reminder by $R_{1,b}$

$$R_{1,b} = \sum_{\nu=1}^n \int_{t_0}^t \int_{t_0}^s \mathcal{L}_{W,\nu}\,\mathbf{f}\,\mathrm{d}W_u^{(\nu)}\,\mathrm{d}s. \tag{42}$$

Applying Fubini's theorem again to change the order of integration, we re-write $R_{1,b}$ in the form of a stochastic integral

$$R_{1,b} = \sum_{\nu=1}^n \int_{t_0}^t (t-u)\,\mathcal{L}_{W,\nu}\,\mathbf{f}\,\mathrm{d}W_u^{(\nu)}. \tag{43}$$

Substituting the operator results in an expression for each dimensional component $i$

$$[R_{1,b}]_i = \sum_{\nu=1}^n \int_{t_0}^t (t-u)\left(\sum_{j=1}^D \frac{\partial f_i}{\partial X^{(j)}} \sigma_{j\nu}\right)\mathrm{d}W_u^{(\nu)}, \quad \text{for } i=1,\ldots,D. \tag{44}$$

In matrix notation, this corresponds to

$$R_{1,b} = \int_{t_0}^t (t-u)\,\mathbf{J}_f\,\boldsymbol{\sigma}\,\mathrm{d}\mathbf{W}_u. \tag{45}$$

The remainder $R_{1,b}$ is a stochastic integral with zero mean, but non-zero covariance, given by

$$\mathrm{Cov}(R_{1,b}) = \langle R_{1,b}\,R_{1,b}^\top \rangle = \int_{t_0}^t (t-u)^2\,\mathbf{J}_f\,\boldsymbol{\sigma}\boldsymbol{\sigma}^\top\,\mathbf{J}_f^\top\,\mathrm{d}u. \tag{46}$$

For sufficiently smooth $\mathbf{J}_f$ and small time step $\tau = t - t_0$, this covariance scales on the order of $\tau^3$.

The term $R_{1,b}$ quantifies the contribution to the remainder arising from stochastic fluctuations of the noise acting through the spatial derivatives of the drift $\mathbf{f}$. It does not contribute to additional systematic bias, but introduces variance in the drift estimator, especially when $\boldsymbol{\sigma}$ or $\mathbf{J}_f$ are large.

## B.3   THIRD REMAINDER TERM $R_{1,c}$

We denote the third remainder term by $R_{1,c}$ and re-write here for convenience

$$R_{1,c} = \int_{t_0}^{t} \int_{t_0}^{s} \mathcal{L}_u^{\dagger} \boldsymbol{\sigma}(\mathbf{X}_u, u) \, \mathrm{d}u \, \mathrm{d}\mathbf{W}_s. \tag{47}$$

In the general case of time- and state- dependent diffusion the integrand of this term would be expressed for the $i$-th row and $\ell$-th column component of $\boldsymbol{\sigma}$ as follows

$$\left[\mathcal{L}_u^{\dagger} \boldsymbol{\sigma}(\mathbf{X}_u, u)\right]_{i\ell} = \frac{\partial}{\partial u} \sigma_{i\ell}(\mathbf{X}_u, u) + \sum_{j=1}^{D} \frac{\partial \sigma_{i\ell}}{\partial X^{(j)}}(\mathbf{X}_u, u) f_j(\mathbf{X}_u, u) \tag{48}$$

$$+ \frac{1}{2} \sum_{j,k=1}^{D} \frac{\partial^2 \sigma_{i\ell}}{\partial X^{(j)} \partial X^{(k)}}(\mathbf{X}_u, u) [\boldsymbol{\sigma}\boldsymbol{\sigma}^{\top}]_{jk}(\mathbf{X}_u, u). \tag{49}$$

However, in our setting we consider state- and time-independent diffusion matrix, and thus $\mathcal{L}_u^{\dagger} \boldsymbol{\sigma}(\mathbf{X}_u, u) = \mathbf{0}$, and by consequence $R_{1,c} = \mathbf{0}$

### B.3.1   FOURTH REMAINDER TERM $R_{1,d}$

The fourth remainder term is

$$R_{1,d} = \sum_{\nu=1}^{n} \int_{t_0}^{t} \int_{t_0}^{s} \mathcal{L}_{W,\nu} \, \boldsymbol{\sigma} \, \mathrm{d}W_u^{(\nu)} \, \mathrm{d}\mathbf{W}_s. \tag{50}$$

For each component $(i, \ell)$ of $\boldsymbol{\sigma}$

$$[\mathcal{L}_{W,\nu}\boldsymbol{\sigma}]_{i\ell} = \sum_{j=1}^{D} \frac{\partial \sigma_{i\ell}}{\partial X^{(j)}} \, \sigma_{j\nu} \, = \, \mathbf{0}. \tag{51}$$

Thus, the omission of this remainder term does not contribute any bias or variance to the EuM-based drift estimator.

