# OpenReview forum: "Flow curvature explains failed SDE drift estimation under sparse sampling"
_ICLR.cc/2026/Workshop/GRaM — ICLR 2026 Workshop GRaM Poster_

### Official Review · Reviewer_ueRP · 2026-02-14
**Curvature causes sparse-sampling drift bias; geodesic-guided augmentation reduces it.**

**Rating:** 6
**Confidence:** 4

**Review:**

Nice insight: sparse-sampling drift regression fails mainly because Euler–Maruyama treats the hidden path between samples as straight, while nonlinear flows bend, giving an O(τ²) bias tied to curvature evolution (∇(Jff)·f). Their fix, estimate an invariant-density metric, connect samples with geodesics, and constrain bridge-based path augmentation around those curves, seems to recover drift much better at large τ on the tested Langevin systems. I’d still like clearer evidence on scalability and robustness (density/metric sensitivity, geodesic compute cost, higher-D, and state-dependent diffusion) and stronger baselines beyond standard GP + simple bridges.

**Pmlr Suitability:**

NA

---

### Official Review · Reviewer_torB · 2026-02-19
**Geodesics to reduce drift approximation error under sparse sampling**

**Rating:** 7
**Confidence:** 4

**Review:**

The paper is well-written and does a good job of motivating the problem. The idea of mitigating the error introduced by the Euler–Maruyama using geodesics on the data manifold is very interesting. However, I believe the construction of the invariant-density metric requires further justification. While the proposed construction is convenient for its simplicity, a more detailed discussion of alternative metric constructions and their impact on the geodesics would strengthen the work.

There is also potential for both theoretical and experimental extensions. Theoretically,  it would be nice to expand on how geodesic interpolation reduces the approximation error, as well as to analyze the impact of different classes of SDEs on the Euler-Maruyama error.
Experimentally, It would be useful to investigate the computational cost of finding these geodesics, and to ensure convergence in their numerical implementation.

**Pmlr Suitability:**

NA

---

### Official Review · Reviewer_8ju3 · 2026-02-23
**Recommendation: reject**

**Rating:** 3
**Confidence:** 2

**Review:**

Quality:
This reads like a low-quality submission.
- several notations are used before being defined
- Figure 1's caption is unsatisfactory
- Figure 2 is not mentioned in the main text
- the figures feel disconnected from the main text
- most importantly, the empirical validation consists of three sentences. This is insufficient.

Clarity:
- did not write down the SDE in question
- did not define $J_f$ and $f$
- no related work is described
- I do not understand how to interpret Figure 2. It is not referenced anywhere in the text and looks more like a review of related methods

Originality:
Hard to judge, because no related work is provided. But I found several works from 2022-2023 with the some of the same **exact** figures.

Significance:
The problem being solved in not put in any context. Thus, I do not view this as a significant contribution.

**Pmlr Suitability:**

NA

---

### Meta-Review · Area_Chair_UcFe · 2026-02-23

**Decision:**

Accept

**Metareview:**

This paper considers the problem of estimating the drift of an SDE given sparse observations by mitigating the error in Euler-Maruyama using geodesics on the data manifold. Two of the reviews are positive with claims that the paper is well-written and provides nice insight. The third reviewer expresses some concerns mainly with the presentation of the paper. Nonetheless the positive reviews outweigh the negative reviews and I therefore recommend acceptance. However, I also recommend that the authors take the feedback from Reviewer 8ju3 into account to improve the paper.

**Relevance To Proceedings:**

Tiny paper — does not apply

**Relevance To Workshop:**

Yes — suitable for GRaM

---

### Decision · Program_Chairs · 2026-03-02

Accept (Poster)